# Cancer Nanomedicine Special Issue Review Anticancer Drug Delivery with Nanoparticles: Extracellular Vesicles or Synthetic Nanobeads as Therapeutic Tools for Conventional Treatment or Immunotherapy

**DOI:** 10.3390/cancers12071886

**Published:** 2020-07-13

**Authors:** Maria Raffaella Zocchi, Francesca Tosetti, Roberto Benelli, Alessandro Poggi

**Affiliations:** 1Division of Immunology Transplants and Infectious Diseases, IRCCS San Raffaele Scientific Institute, 20132 Milan, Italy; zocchi.maria@hsr.it; 2Molecular Oncology and Angiogenesis Unit, IRCCS Ospedale Policlinico San Martino, 16132 Genoa, Italy; francesca.tosetti@hsanmartino.it (F.T.); roberto.benelli@hsanmartino.it (R.B.)

**Keywords:** ADAM10, aminobisphosphonates, polymeric nanoparticles, exosomes, zoledronic acid, alendronic acid

## Abstract

Both natural and synthetic nanoparticles have been proposed as drug carriers in cancer treatment, since they can increase drug accumulation in target tissues, optimizing the therapeutic effect. As an example, extracellular vesicles (EV), including exosomes (Exo), can become drug vehicles through endogenous or exogenous loading, amplifying the anticancer effects at the tumor site. In turn, synthetic nanoparticles (NP) can carry therapeutic molecules inside their core, improving solubility and stability, preventing degradation, and controlling their release. In this review, we summarize the recent advances in nanotechnology applied for theranostic use, distinguishing between passive and active targeting of these vehicles. In addition, examples of these models are reported: EV as transporters of conventional anticancer drugs; Exo or NP as carriers of small molecules that induce an anti-tumor immune response. Finally, we focus on two types of nanoparticles used to stimulate an anticancer immune response: Exo carried with A Disintegrin And Metalloprotease-10 inhibitors and NP loaded with aminobisphosphonates. The former would reduce the release of decoy ligands that impair tumor cell recognition, while the latter would activate the peculiar anti-tumor response exerted by γδ T cells, creating a bridge between innate and adaptive immunity.

## 1. Introduction

Nanotechnology development evolved very quickly in recent years, offering more than an option in the biomedical field and providing a variety of tools for diagnosis and therapy of several diseases [1]. Different combinatorial nanoparticles (NP) have been designed and tested for cancer imaging and therapy, due to their properties that allow selective tissue localization, targeting and drug delivery. Other reasons why NP are attractive for nanomedicine rely on the reduction of systemic toxicity, maintenance of therapeutic efficacy, greater safety and biocompatibility, increased solubility, higher stability and faster delivery. NP are defined as particles below 100 nm of dimension; although their surface is generally large enough to bind and carry diagnostic or therapeutic compounds, some drugs need relatively larger NP to reach optimal delivery rate [2,3]. In addition, biomaterials used to assembly NP are of striking importance: the composition of NP may be of biological origin, like lipids, lactic acid, dextran, or chitosan, or chemicals, such as carbon, silica, and many polymers.

The intrinsic characteristics of NP make them suitable for integrated diagnosis and therapy of cancer, including visualization and quantification of tumors at the site of lesion and biodistribution of carried drugs, leading to personalized nanomedicine, as well [4,5]. Molecular nanoprobes and contrast agents have been applied for non-invasive imaging of tumors; among them, ultra-small superparamagnetic iron oxide NP and gold NP, coupled to low molecular weight contrast agents, have been used for magnetic resonance or computerized tomography [4,5]. However, there are still limitations in the use of NP for radiodiagnostic purposes: so far, there are a few nanoformulations, namely iron oxide NP, used in clinical practice: ferucarbotran (Resovist, Bayer Schering Pharma, Berlin, Germany) and ferumoxytol (Feraheme; AMAG Pharmaceuticals Inc., Cambridge, MA, USA), approved by FDA (Food and Drug Administration), for the treatment of anemia, are now used off-label in radiodiagnosis [6,7].

NP for drug delivery are generally considered carriers of pharmaceutically active compounds, although nanoformulations of the drug itself have been described [3,8]. In any case, major points to be faced are drug incorporation rate, stability and half-life, NP biocompatibility, distribution and drug delivery rate [9]. The net result of optimal NP drug formulation would be the enhanced delivery to the site of lesion and reduced or absent toxicity for bystander or non-target tissues, that is, an increase of the therapeutic index. In principle, this would mean that the best NP should be long-lived and target-specific, although short life span is preferable when rapid achievement of high doses is required.

A non-negligible problem in the use of drug nanomaterials is the entrapment in the phagocytic system [10,11]. Surface modifications or coating with polyethylene glycol (PEG) or poly(vinylpyrrolidone) (PVP) (see Section 4.1) would contribute overcoming this problem, mainly preventing or inhibiting phagocytosis [12,13]. These modifications also prevent agglomeration and reduce their potential toxicity and; on the other hand, they have higher local therapeutic concentrations [14]. The major side effects are mainly due to the ability of NP to induce inflammation, and this is related not only to the dimension but also to the biomaterial [10,11]. In addition, pro-thrombotic effects, due to the interaction with microvessel endothelial surface, have been reported [2].

So far, several therapeutic NP have been applied in clinical practice. Antiblastic drugs, such as PEGylated, doxorubicin-loaded liposomes, paclitaxel containing albumin NP or liposomal amphotericin B, and other nanoformulations are currently tested in preclinical and clinical trials [15,16]. Extracellular vesicles (EV) and exosomes (Exo), derived from cellular budding in normal and tumor cells, have been exploited as “natural” NP to carry anticancer drugs fora very long time [17,18]. Among EV, Exo are more appropriate NP due to their size, ranging between 30 and 150 nm, which allows them to spread into the extracellular microenvironment, reach neighboring cells, and interact with their cell membrane [19]. This makes Exo physiological transporters of factors and potential carriers of drugs easily deliverable into target cells. A more recent nanotechnology application is cancer immunotherapy: agents used to activate and boost the immune system, such as cytokines or monoclonal antibodies (mAbs) for checkpoint blockade, have been included in nanotechnology platforms to improve efficacy and overcome immune evasion [20].

This review provides a brief overview of the recent advances in nanotechnology applied for theranostic use. In particular, we will describe the main biological or chemical nanosystems proposed for cancer immunosurveillance and treatment, including new possibilities for anti-tumor drug delivery and immunotherapy.

## 2. EV as Drug Delivery Vehicles in Cancer

### 2.1. Extracellular Vesicles in the Physiopathology of Intercellular Communication

The evidence that extracellular vesicles (EV) released from different tissues preserve the fingerprint of the cell of origin [18,19] is a feature functional to EV application for diagnostic and prognostic purposes, recurrently proposed in recent clinical trials. The knowledge emerged in more than 20 years of investigations, highlighting a role for EV in energy-demanding processes related to tissue trophism and regeneration, where cellular metabolism and cell communication at a distance are indissolubly intertwined [20,21,22,23,24]. For example, allogenic EV derived from mesenchymal stem/stromal cells (MSC) show potent regenerative activity in target tissues [22,23,24,25], and this property is exploited in clinical trials for tissue repair [26]. Increased EV release in cell stress, hypoxia, tissue acidosis, and cancer [27] also tells us about EV belonging to an emergency alerting system [28], which has to be switched off periodically in homeostatic conditions, following circadian rhythms, as well [29]. In cancer, EV-mediated processes result in stemness maintenance and establishment of a premetastatic niche and complex remodeling events, such as epithelial-mesenchymal transition (EMT) [30], angiogenesis [31], and metastatic invasion [32]. Tumor-derived EV also drive a malignant modulation of the local microenvironment [27,28] and drive the establishment of organ-specific metastases [33]. On the other side, stromal cell-derived EV can increase tumor transformation and resistance to drug therapy [30,31,32,33,34,35,36,37]. In this perspective, tumor-derived EV represent a negative marker for patient prognosis [36,37,38,39].

Despite their negative role in cancer progression, EV can also be employed as valuable carriers for therapeutics. EV conversion into anti-tumor drug vectors represents a challenging goal. Among limiting factors, it is the feasibility of purifying significant amounts of therapeutic, good manufacturing practices (GMP)-grade EV and the need to set appropriate intermediate endpoints to measure bioavailability and efficacy of EV-conveyed drugs. Thus, the translational application of EV as drug carriers seems difficult for several still open questions [19,39]. Nevertheless, some studies explored the use of EV for drug delivery in cancer (also see next Section) and degenerative diseases [19,36,39,40].

### 2.2. EV Sources and Loading

Tumor cell lines have been the principal source of EV for in vitro studies in the past, as they can be easily transfected with desired constructs, or loaded with cytotoxic drugs, and expanded in large quantities. Moreover, their EV usually exhibit a natural tropism towards the parental cell line in vivo, optimizing targeting in mouse models [41]. Tumor-derived EV also contain pathologic signals, like oncogenic molecules; thus, their transfer to the clinic is improbable, despite preliminary applications on cancer patients confirmed their therapeutic potential in particular settings [42]. Indeed, tumor-derived EV are mainly studied as prognostic and predictive markers for cancer patient monitoring.

Normal human primary cell cultures, like MSC and dendritic cells (DC), could represent a less dangerous source of EV. MSC have been used in regenerative medicine due to their ability to differentiate into several tissues; moreover, they show anti-inflammatory and immunosuppressive effects, increasing tolerance after infusion [35]. EV from bone marrow-derived MSC showed a direct therapeutic effect on HEP-G2, Kaposi’s sarcoma, and Skov-3 cell lines, both in vitro and in vivo [43]. In vitro, paclitaxel-loaded MSC-derived EV showed strong inhibition of CFPAC-1 pancreatic tumor cell line, with a 50% reduction of tumor growth in vivo [40]. As reported before, MSC could also carry molecules favorable to the tumor, representing a dangerous double-edged sword. Although, in multiple myeloma (MM), only EV purified from patients’ MSC showed a tumor-promoting activity on MM cells, while EV from the MSC of healthy subjects reduced MM growth both in vitro and in vivo [44]. According to these studies, MSC from healthy individuals show sufficient safety and intrinsic anti-tumor activity, sustaining their use as reliable sources of EV.

DC-derived EV have been tested as immune modulators and applied to cancer therapy in phase I–II trials [45]. Unpredictably, this approach showed mild effects on T-cell activation, while an increased natural killer (NK) cell response was observed. The use of EV for immune-driven anticancer responses will be discussed in the next chapter.

Plasma and milk represent two additional sources of EV. Autologous EV can be purified in large quantities from blood plasma, while cow milk is an inexpensive source of allogeneic EV. Plasma-derived EV composition reflects the age and the health status of the donor [41,42,43,44,46,47,48,49], thus limiting the use of plasmatic EV in the autologous setting. These EV sources yet undergo pioneering studies, testing purification and loading techniques; thus, their transfer to the clinic for cancer therapy is immature.

To achieve therapeutic purposes, EV loading of specific therapeutic molecules can be obtained by manipulating isolated EV (exogenous loading) or carrying the desired agent onto parental cells (endogenous loading) (Figure 1). The former applies to EV sources not relying on a cultured cell type that should release the drug-loaded EV, like plasma. Once a sufficient amount of GMP-grade purified EV is obtained, this method is reproducible and broadly applicable. Lipophilic molecules like paclitaxel and curcumin can directly permeate the lipid bilayer [50,51], while small interfering RNA (siRNA) can be made hydrophobic and complexed to cholesterol for a passive transfer in EV [52]. In turn, endogenous loading allows the direct use of EV/Exo-producing cells to carry modified therapeutic agents [53,54]. While the direct use of drugs on living cells has the intrinsic limit of drug toxicity, whichcan reduce the yield of EV [55], the engineering of producing cells with therapeutic expression vectors is a reliable strategy. For example, EV secreted by MSC transduced with lentivirus expressing human TNF-related apoptosis-inducing ligand (TRAIL) have been suggested to induce cancer cell death as an alternative to recombinant TRAIL that shows poor pharmacodynamics [56]. Recently, this approach has been proposed for targeted delivery of oncolytic adenoviruses or therapeutic miRNA to cancer cells; in this model, the molecule of interest is transfected/infected to a cell line in vitro, and EV are then purified and used as therapeutic tools [57,58,59].

EV targeting remains the most important challenge to fight tumors. In animal models, when EV are injected in the bloodstream, they rapidly accumulate in the liver, spleen and lungs by macrophages uptake, and are degraded within 6 h [60,61]. A localized injection of EV can obtain different distributions, enriching the gastrointestinal tract, regional lymph nodes, or central nervous system [62,63]. EV surface is highly glycosylated, and O-glycan residues retard EV uptake by the lung but not by the liver and spleen. Accordingly, increased lung targeting can be simply obtained by EV deglycosylation [64]. On the other side, unspecific protease pre-incubation causes a prolonged half-life in the bloodstream and a reduced uptake of EV by lungs [65]. Both protease digestion and PEG coating, while prolonging the half-life of EV, represent a step-back for the rationale use of EV as substitutes of liposomes, overriding the protein pattern exposed on EV surface and limiting EV-cell interaction [66]. Moreover, functionalization with amino-ethylanisamide targeted the sigma receptor expressed by lung cancer cells [67]. Thus, the specific engineering of EV surface remains a more promising strategy for tumor targeting.

### 2.3. Further EV Modifications for Therapeutic Purposes

A simple approach to increase the specific targeting of tumor cells by EV is their enrichment with hyaluronic acid (HA) conjugated to lipid chains. The increased HA content of EV surface allowed the uptake of doxorubicin-loaded EV by CD44 expressing tumors, with increased efficacy compared to wild type EV [68]. Alternatively, bridging molecules can be used to create new EV-tumor interactions: glycoprotein A33 is specifically expressed in the gut epithelium and colorectal carcinomas (CRC) [69]. The use of iron-oxide nanoparticles coated with the A33 antibody allowed the immediate targeting of doxorubicin-loaded, colon cancer cell line-derived EV to colon tumors [70]. More sophisticated approaches can fuse targeting proteins to proteins expressed on EV surface. Lactadherin/MFGE8 localizes into EV membrane by tight binding of its C1C2 domain with phosphatidylserine [71]. Chimeric proteins carrying the C1C2 domain can be produced in bulk and incorporated into EV from different sources after purification. This approach has been used to display epidermal growth factor receptor (EGFR)-binding nanobodies on EV surface [72]; these EV bound with high affinity to A431 cells expressing EGFR but not to EGFR-negative Neuro2A cells. Using a similar approach the C1C2 domain was fused to the single-chain variable fragment of the anti-human epidermal growth factor receptor (HER)2 antibody ML39 and expressed in HEK293 cells [73]. The same cells were transfected with a construct for the expression and accumulation in EV of a bacterial mRNA whose product can convert a pro-drug into its active metabolite. The resulting EV were used to target in vitro and in vivo BT474 breast cancer cells, carrying HER amplification, in the presence of the prodrug. This approach was able to inhibit the growth of BT474 tumors in vivo, reducing the systemic toxicity of the drug. Lysosome-associated membrane glycoprotein 2b (Lamp2b) is a valid substitute of the C1C2 domain, as it is expressed in exosomes. The chimera between the iRGD peptide from alpha-v integrin subunit and Lamp2b was used to target doxorubicin-loaded EV against MDA-MB-231 breast cancer in vitro and in vivo, suppressing tumor growth [74]. An interleukin (IL) 3-Lamp2b chimera was used to target chronic myeloid leukemia cell lines with Imatinib, or B cell receptor-Abelson (BCR-ABL) siRNA-loaded EV [75]. Both strategies showed specific interaction of EV with chronic myelogeneous leukemia (CML) cells expressing IL3 receptor (R) and a strong reduction of CML tumor growth in vivo. In addition, RNA aptamers can be used to target cell-specific ligands, when exposed on the outer surface of EV. For example, RNA aptamers, directed to prostate-specific membrane antigen or EGFR, were fused with the bacteriophage phi29 pRNA and a cholesterol tail; these engineered-EV, loaded with a surviving-specific silencing RNA, could inhibit prostate or breast tumors growth in vivo [76].

This is only a limited review of the numerous strategies that have been recently developed to obtain an optimized targeting of cancer by engineered EV, although the transfer of these techniques to the clinic is apparently at a stalemate [77]. At the date of this review, the clinical trial database (https://clinicaltrials.gov/) reported 83 records when the term “exosome” (Exo) was searched for, in any cancer-related study. Although, most studies investigated tumor or circulating EV, looking for prognostic/predictive/diagnostic markers, while therapeutic applications were rare. Three notable exceptions are protocols NCT01294072, a phase I trial started in 2011 using grapefruit-derived Exo to deliver curcumin to gut mucosa and colon tumors; NCT01159288, a phase II trial testing lung cancer vaccination with DC-derived Exo loaded with tumor antigen; NCT03608631, a recent phase I trial aiming to test MSC-derived Exo loaded with KrasG12D siRNA against KrasG12D-positive metastatic pancreatic cancer. The main characteristics and therapeutic uses of EV and Exo are summarized in Table 1.

## 3. Exo as Drug Carriers to Enhance Anti-Tumor Immune Response

### 3.1. EV and Exo in Tumor Immunity

Among EV, Exo represent the smallest vesicles (30 to 150 nm) that originate from the cell endosomal compartment in multivesicular bodies and are delivered into the extracellular space [19]. Like all EV, Exo contain cytoplasmic or membrane molecules, derived from the budding process, which can interact with bystander cells and deliver signals with positive or negative effects on tumor cell growth, leading to intense intercellular communications in the tumor microenvironment (TME). Since early pioneering studies EV and Exo appeared as important tools with clinical potential [72,73]. The actual state of the art of EV and Exo applications as such or for drug delivery in the clinics can be estimated by active clinical trials employing EV for therapeutic use (also see Section 2.3).

EV have been characterized by their pro-inflammatory and immunosuppressive activity in the TME [77,78]. In Hodgkin lymphoma (HL) tumor-derived EV and Exo induce a pro-inflammatory phenotype in fibroblasts, causing a crossfire effect [79]. Different pharmacologic strategies are in progress to educate stromal cells, tumor macrophages, and immune cells to revert to immunostimulatory phenotypes [80] and turn circulating EV into anti-tumor vehicles.

A relevant application of EV in preclinical studies exploits a set of immunomodulatory molecules, now recognized as specific components of different subpopulations of tumor EV as drug targets [81]. Accumulating evidence indicates that EV carry a considerable pool of cell receptors, i.e., programmed death receptor ligand (PDL)1 and EGFR [82], which can compete for binding to therapeutic molecules. Moreover, exosomal PDL1 [83] and EGFR induce an immunosuppressive microenvironment [82].

EV-bound therapeutics could be considered as sequestered molecules or a circulating fraction to be delivered to target cells. The amount of this circulating pool of EV-bound receptors can increase or be blocked under pharmacologic stimulation. Experimentally, this has been observed by using modulators of the exocytotic pathway; indeed, EV biogenesis is decreased by the inhibitor of neutral sphingomyelinase GW4869 or nocodazole and increased by lysosomal inhibitors. Of concern, EV can be modulated by common drugs (simvastatin) [84], chemotherapy agents (cyclophosphamide), and tumor hypoxia [85], or the acidic TME established by lactate [86]. Thus, some pharmacological treatments could create an unwanted off-target immune imbalance through EV with negative consequences on the immune response.

Circulating tumor or stromal receptors carried by EV/Exo, besides cell resident receptors, can play a relevant role in immunoregulation [87,88,89,90,91,92,93,94]. In non-Hodgkin lymphomas (NHL), for example, B-cell EV carrying CD20 on their surface bind to the anti-CD20 antibody rituximab shielding target cells from antibody-dependent cellular cytotoxicity (ADCC) [87]. Treatment strategies to overcome this model of rituximab resistance point to the blocking of EV secretion through the use of indometacin, which downregulates at the transcriptional level the expression of the adenosine triphosphate binding cassette (ABC) transporter ABC subfamily member A3, or rapamycin, which alters multivesicular bodies biogenesis [87].

In the case of HL, EV/Exo express higher levels of CD30 than other membrane receptors, therefore circulating CD30 in EV/Exo received special attention due to the relevance of CD30 in HL diagnosis and the effectiveness of anti-CD30 mAb brentuximab-vedotin in HL therapy [88,91,94,95].

Among the strategies to improve tumor surveillance, engagement of natural killer (NK) activating receptors, such as NKp30 and the natural killer group 2D receptor (NKG2D), plays a prominent role due to NK extended lifespan and the evidence of NK immune memory [94,95,96]. Further, a valuable NKG2D feature is it sbinding to functionally redundant major histocompatibility complex (MHC) class I-like self-antigens (MICA) usually expressed at low levels in normal cells and overexpressed on tumor cells [91]. As mentioned, over-activation of membrane shedding of regulatory molecules and receptors in tumors can exert immunosuppressive effects. Indeed, soluble ectodomain fragments can compete for drug binding of the cell membrane-associated molecules [92,97,98,99]. Shedding of NKG2D ligands is one of the mechanisms responsible for tumor immune evasion [91,93,94,95,98,99,100]. A disintegrin and metalloproteinase (ADAM) 10 endopeptidase is a dominant cell transmembrane sheddase acting in concert with other proteases to cleave substrates involved in tumor immune activation and immune escape [94,101]. In HL, ADAM10 is responsible for the shedding of both CD30 [79] and the majority of NKG2D ligands [95].

Based on these data and considerations, ADAM10 is recognized as an attractive drug target to improve anti-tumor immunity, a topic extensively reviewed in recent papers [96].

### 3.2. Exosomal ADAM10 and ADAM10 Inhibitors in Cancer

ADAM10 is present in Exo from various tissues and appears in the exosomal markers list, together with the tetraspannins CD9, CD81, and CD63 [97]. Notably, the association of ADAM10 to the C8 family of tetraspannins has been identified as a mechanism to achieve specificity of action [98]. However, the relevance of the differential sorting and specific cell targeting of exosomal ADAM10 remains to be defined [99]. ADAM10 overexpression, reported in different tumor models, probably results from resident cancer epithelial cells, MSC and EV released into the extracellular space. Given that ADAM10 shedding of immune mediators results in an immunosuppressive microenvironment [96], it is conceivable that exosomal ADAM10 can favor immune privilege in the TME through enhanced shedding of substrates from tumor cells, including NKGD2 ligands, which nullifies NK cell-mediated effector functions [94]. Thus, exosomal ADAM10 as a druggable target could serve to boost delivery of inhibitors, as well as to enhance tumor immunity. Hence, the advantage of developing ADAM10 specific inhibitors.

One of the first ADAM10 inhibitor, GI254023X, was found by the screening of hydroxamate compounds inhibiting recombinant metalloproteinases [100]. GI254023X was 100-fold more potent in inhibiting ADAM10 compared to ADAM17. Despite its specificity, GI254023X was notfurther developed for clinical settings and is commercially available for experimental purposes.

At present, the clinical trial database (https://clinicaltrials.gov/) reported 7 records for the search term “ADAM10”. In particular, a phase I multicenter clinical trial is testing the ADAM10/17 inhibitor INCB7839 (Aderbasib, Incyte Corporation, Wilmington, DE, USA) in recurrent or progressive gliomas while a phase I/II study, as consolidation therapy, in diffuse large B cell lymphoma patients is currently under investigation (NCT02141451). In the literature, ADAM10 inhibitors have been used in metastatic breast cancer xenografts [101], while two clinical trials with Aderbasib in breast cancers have been terminated as the development of this compound has been suspended. In this context, we focused our attention on novel and more specific ADAM10 inhibitors with potential clinical value to balance immune escape by exosomal delivery to tumor tissues. Rossello and coworkers synthesized ADAM10 inhibitors (MN8, LT4, and CAM29) with good hydrophilicity which are taken up by cells upon binding to membrane surface ADAM10 and enter the endolysosomal compartments [102]. These inhibitors and their fluorescent counterparts were used to study ADAM10 intracellular distribution and to explore their release into the extracellular space bound to ADAM10 in EV [88]. Based on experimental evidence [103], we reasoned that the drug cargo delivered through Exo in subcellular locations could better reach the intracellular ADAM10 pool, otherwise inaccessible to hydrophilic molecules, achieving ADAM10 blockade before its insertion into the plasma membrane. Notably, EV isolated from MSC of HL patients and HL tumor cell lines treated with the specific ADAM10 inhibitors (Figure 2) retain the ability to inhibit shedding of the ADAM10 substrates MICA, tumor necrosis factor (TNF)α and CD30 in recipient cells [88]. Indeed, purified fluorescent EV obtained from HL cells treated with fluorescein isothiocyanate (FITC)- or cyanine (CY)5-conjugated inhibitors enter MSC, and vice versa, as fluorescent inhibitors carried by vesicular ADAM10 could be traced in intracellular vesicular compartments upon uptake [88]. ADAM10 inhibitors were also able to preserve or enhance HL tumor cell ADCC triggered by anti-CD30 mAb iratumumab, thus showing pleiotropic beneficial effects [88].

Thus, the release into the HL microenvironment of Exo-like microvesicles carrying ADAM10 can result in the shedding of cytokines, including TNFα, which work as a lymphoma growth factor, or soluble molecules, such as sMICA and sCD30, which interfere with host immune response and with antibody immunotherapy. EV containing ADAM10 blockers would interfere with this process, allowing both the anti-lymphoma immune response and the humanized mAb-immunotherapy [88].

The results described raised the question of possible enzymatic activity of intracellular ADAM10, as reported by others [104]. We obtained preliminary evidence that HL tumor cells and MSC Exo can actively process recombinant or endogenous ADAM10 targets, such as TNFα and FasL, in in vitro assays with intact Exo; this indicates that some substrates are accessible to the ADAM10 carried by purified Exo, whatever its orientation inside these vesicles. In our HL models, ADAM10 substrate processing was inhibited in the presence of Exo isolated from HL or stromal cells primed with ADAM10 inhibitors. In addition, ADAM10 inhibitors LT4 and MN8 [102] induce a profound reorganization of endolysosomal trafficking and membrane dynamics (unpublished observations), raising several unsolved questions that will need further studies to be addressed.

## 4. NP in Anti-Cancer Theranostic Strategies

### 4.1. NP Types

Among the huge number of NP developed so far, we will focus on the main types of nanovectors used as drug carriers in cancer therapy. Each type shows advantages and disadvantages, related to biocompatibility, half-life, and distribution, drug delivery rate, and toxicity so that the optimal formulation is still to be defined [105,106]. First, nanodrugs are made of two components: nanovector and drug, which may interact with each other, or they may not. Second, NP can be divided into two groups, based on the biomaterial composition and biodynamic properties: natural biodegradable and synthetic engineered (Figure 3).

Liposomes are an example of natural NP, with an aqueous core that can load hydrophilic molecules, surrounded by phospholipids and a cholesterol bilayer. Liposomes are more biocompatible than other NP and protect the drug from degradation; however, due to this latter characteristic, it is not always easy to control bio-distribution and toxicityof the carried drug [105,107]. Liposomes can also modify anticancer agents as reported by Liu and coworkers that, using an alanine-alanine-asparagine-trans-activating factor (AAN-TAT-liposome platform, made a doxorubicin carrier able to enhance the tumoricidal effect and reduce side effects [108].

Polymersare subdivided into natural polymers, including proteins, peptides, glycans, and cellulose, or synthetic polymers, derived from natural monomers, such as polylactic acid (PLA) and poly (lactic-co-glycolic acid) (PLGA). A particular type is represented by microbial fermentation polymers, such as poly-hydroxybutyrate [105]. Natural polymers used in NP composition include chitosan, dextran, albumin, heparin, gelatin, and collagen [109,110]. PLGA and chitosan NP are good vehicles for the transport and delivery of proteins [111,112,113]. Thermosensitive polymers, triggered by temperature variations, have been used to control drug delivery [111,114].

Dendrimers are globular macromolecules with a central core, branches called “generations”, repeat spherical units, and terminal functional groups (Figure 3). Due to this architecture, dendrimers can be exploited to carry molecules on the branches, through the peripheral functional groups, or in the internal molecular spaces [115,116,117]. In addition, they are self-assembled and can be linked to other NP, such as liposomes, to modulate their solubility and drug delivery [118,119]. Dendrimers have been used as gene vectors or to produce contrast agents for molecular imaging [109,120].

Micelles are composed of an internal core and an external shell that can be hydrophilic, often represented by PEG, or hydrophobic when PLA or PLGA are used [111,112,121]. These NP display good biocompatibility, as well as the ability to carry either hydrophilic or hydrophobic drug, allowing the solubilization and bioavailability of undissolvable drugs, such as paclitaxel and docetaxel. PEG-PLA polymeric micelle preparation loaded with paclitaxel (Genexol-PM), showing reduced toxic effect compared to other drug formulations, was approved by the FDA for the treatment of non-small cells lung cancer [122]. Micellar NP have been widely used for passive targeted cancer therapy exploiting modifications of molecules at their surface, binding copolymers, or inserting thermo-or light-sensitive groups in their structure [105].

Inorganic nanoparticles, such as quantum dots, superparamagnetic iron oxides, gold nanoparticles, and carbon nanotubes, have been used to improve the efficiency of tumor imaging and radiotherapy [109,123]. Some of these NP can be subjected to surface modifications, such as binding of antibodies or ligands (see Section 4.2), making them capable of specific receptor-targeting, or act as magnetic resonance imaging contrast agents. Quantum dots and gold nanoparticles display intrinsic optical, electrical, and magnetic properties [124] useful for intracellular imaging. In early clinical trials, some inorganic nanomaterials, such as gold nanoparticles [125] and silica nanoparticles [126], have shown toxicity and reduced stability. Thus, NanoTherm [127] is the only formulation approved for clinical use in glioblastoma that can be thermally ablated by magnetic hyperthermia induced by superparamagnetic iron oxides. The main features, advantages, and disadvantages of the above-mentioned NP are listed in Table 2.

### 4.2. NP Surface Modifications for Passive or Active Targeting

Passive targeting defines the mean favoring the accumulation of a drug at the tumor site and is dependent on the ability of the drug carrier to pass vascular endothelium and biological barriers to reach the tissue of delivery (Figure 3A). These properties are directly related to the size of the particle used as a vehicle and to the route of administration [107]. In the case of oral administration, particles less than 5–10 µm are suitable to cross the intestinal mucosa, whereas, for intravenous infusion, NP are indicated, since capillaries diameter can be smaller than 5 µm. Moreover, particles larger than 100–150 nm are easily cleared or entrapped by the phagocytic reticuloendothelial system [128,129]. On the other side, particles smaller than 10–20 nm are rapidly washed out by the kidney or can aggregate and provoke embolisms [130,131]. Thus, the best NP size ranges between 50 and 100 nm [132,133].These NP were reported to exploit the fenestrated endothelium of tumor vessels through the so-called “enhanced permeation and retention effect” (EPR), caused by inefficient lymphatic drainage in tumor tissues [133,134,135]. The consequent high interstitial pressure would lead to enhanced retention of NP in the tumor; due to this effect, macromolecules can be transported and delivered to the tumor, enhancing drug local concentration and efficacy [135,136,137,138,139,140].

Active targeting aims to increase both circulation half-life and selectivity of cargo delivery from NP (Figure 3B). As EV, NP conjugated with PEG increase solubility, size, pharmacokinetics and pharmacodynamics of the nanoformulation [141,142]. Several PEGylated protein drugs, summarized in a review by Morales-Cruz [107], have been approved by FDA for anti-cancer treatment, including PEG-L-asparaginase (Oncaspar^®^), PEG-interferon (PEGIntron^®^), and PEG-granulocyte colony-stimulating factor (Neulasta^®^).

Active tumor targeting, however, needs a more specific localization of the drug to the tumor site than that obtainable exploiting EPR effect or PEGylation. Proteins, including, albumin, transferrin and lectins have been used to this purpose; sugars, small molecules and peptides have also been employed (Figure 3B). The albumin-based protein-bound paclitaxel Abraxane has been approved by the FDA for the treatment of breast cancer, non-small cell lung cancer (NSCLC) and pancreatic cancer showing stability, efficacy, and reduced toxicity towards the soluble drug [143]. BIND-014 is the first PEG-PLGA targeted NP-docetaxel recently reported in phase I/II studies for the treatment of metastatic cancer and KRAS-positive or NSCLC [144]. Among other proteins, also transferrin can serve to specifically drive anti-cancer drugs to the tumor site due to the elevated levels of transferrin receptor expressed by lung, ovarian, colon, and brain cancers [145]. Lectins are proteins that preferentially bind carbohydrates attached to glycoproteins. Carbohydrate-lectin binding can be very useful to target cancer cells since it can be as specific as the interaction between antibody and antigen or enzyme and substrate [146]. Cancer cell surface glycoproteins are often different from those of normal cells, such as P-glycoprotein, widely expressed in many solid cancers. Targeting P-glycoprotein could also ameliorate the response of tumors to therapy, as it is one of the mediators responsible for multidrug resistance syndrome [147,148]. This strategy, based on the use of lectin-based targeted modification of a drug carrier, has recently been reported in pancreatic ductal adenocarcinoma [149].

Folate (FT) is one of the small-molecules used in active, nanoformulated drugs targeting tumors [150]. Indeed, the α and β isoforms of its receptor (FR) are overexpressed in most solid cancers [151]. As an example, nanoformulated cytochrome-c modified with FT has been shown to target FAR-positive HeLa cells, inducing internalization and cytotoxicity, confirming efficacy also in vivo [152].

Among sugars, the linear polysaccharide HA is of considerable interest for active targeting. Indeed, HA is the main ligand for the CD44 receptor, a cell surface molecule overexpressed in a variety of tumors, including ovarian and CRC, besides acute leukemia, so that it is recognized as a tumor marker, in particular, to identify cancer stem cells [153,154]. Thus, HA is of great interest since drugs able to reach and eliminate cancer stem cells are still missing. Of note, HA nanoformulation has a negative surface charge, which prevents the clearance by the reticulo-endothelial system [155]. HA has been linked to different drugs: for example, paclitaxel showed increased efficacy and reduced toxicity when encapsulated in HA NP, and HA-coated NP have been proposed to deliver CD44 siRNA; also, a bioconjugate, which links irinotecan and HA, proved to be effective in vitro and xenograft models [153,156]. Another approach to target CD44 is represented by peptides sharing sequence homology with the HA-binding domain of CD44 [157]. In addition, peptides containing sequences recognized by molecules involved in adhesion and internalization (such as RGD) can be used to assure that drug is taken up efficiently by cancer cells [158,159].

Another recent way of tumor active targeting is represented by the antibody-drug conjugates (ADC) designed to address receptors expressed by cancer cells through antigen-antibody recognition and obtain specific drug delivery. Examples of this type of active targeting are inotuzumab-ozogamicin, a mAb recognizing CD22-linked to the cytotoxic agent calicheamicin, and brentuximab-vedotin, an anti-CD30 therapeutic mAb conjugated to the antineoplastic agent monomethyl-auristatin. The former has been approved by the FDA for the treatment of acute lymphoblastic leukemia patients, the latter is largely used in HL therapy [160,161]. Based on these therapeutic successes, antibody-conjugated NP has recently emerged as a novel strategy for the specific delivery of chemotherapic agents [162]. Advantages of such NP are the high specificity of targeting and the precision of drug release. Despite limited information on biodistribution and pharmacokinetics or toxicity of these nanoformulations, they are attractive and might represent the next future of active targeting in cancer therapy.

### 4.3. Nanoformulations of Drugs to Increase Anti-Tumor Immune Response

The use of nanoformulated drugs has been widely applied in pre-clinical and clinical studies in several kinds of tumors, with promising results [163,164,165,166,167,168]. In the large majority of these studies, cytotoxic drugs have been delivered by nanoformulations to potentiate their anti-tumor effect reducing undesired damage of healthy tissues [163,164,165,166,167,168]. Interestingly, this delivery system can lead to myeloid immune cell stimulation and subsequent release of cytokines, including TNFα and granulocyte macrophage-colony-stimulating factor (GM-CSF), with direct or indirect anti-tumor effects [169]. Nanoformulated recombinant GM-CSF has been engineered for cancer immunotherapy to enhance the clearance of tumor apoptotic cells by macrophages [170]. Along this line, nanoformulations made of a scavenger receptor B type-1 targeting peptide linked with an M2 macrophage binding peptide have been prepared and loaded with a small interfering RNA (siRNA) to target M2 tumor-associated macrophages in melanoma [171]. Targeted gene and RNA-interference delivery by NP have been widely proposed for cancer immunotherapy, both to silence oncogenes and evoke an immune response, although this system is not free of side effects [172,173]. Peptide and protein-based NP are also suggested as tools to improve the efficacy of cancer vaccines [174,175]. On the other side, drug nanoformulations have been used to stimulate anti-tumor immune effector cells [167,168].

The use of specific drugs able to down-regulate the so-called points of control (immune-checkpoint) of the immune response has revolutionized the therapeutic approach to tumors like melanoma and NSCLC. Indeed, the use of humanized mAbs against immune-checkpoint molecules, such as programmed cell death receptor 1 (PD1) and cytotoxic activated lymphocyte molecule 4 (CTLA4), have given exceptional clinical responses in such cancers [176,177]. Unfortunately, several other tumors, such as CRC and ovarian cancer, did not show any benefit from the use of these therapies [177,178,179]. The block of immune-checkpoints can restore the antigen-specific anti-tumor immune response exerted by CD8^+^αβT lymphocytes. Indeed, both PD1 and CTLA4 can downregulate the effector functions exerted by this lymphocyte subset [180,181]. The PDL1 expressed on tumor cells has been used to target liposomal doxorubicin with specific PDL1-binding peptides in a murine model [182]. In this system, the direct anti-tumor effect exerted by the drug was associated with the impairment of the binding between PDL1 and PD1, thus increasing the anti-tumor activity of CD8^+^ T cells [182]. Anti-tumor effector cells of the innate immune system, like NK and γδT cells, express low levels of immune-checkpoint molecules and display remarkable cytotoxic activity against several tumors. Thus, they can be the ideal target of nanoformulated activators for tumors not responding to anti-PD1 and/or anti-CTLA4 therapy [179,183]. In this context, it is important to analyze how γδT cells can be triggered by nanoformulated drugs.

## 5. Nanoformulated Aminobisphosphonates (N-BPs) as Anti-Cancer Immunostimulators

### 5.1. N-BPs as Stimulators of Anti-Tumour γδT Cell Response

Typical anti-tumor effector cells are tumor-specific T lymphocytes expressing the T cell receptor (TCR), composed of the α and β chain heterodimer, coupled to CD8 co-receptor [184,185]. In addition, NK cells are involved in the killing of tumor cells [186,187]. A third lymphocyte population, expressing the γ and δ chains of TCR is relevant in mucosal immunity and shares phenotypic and functional characteristics with both αβ TCR T and NK cells [188,189,190]. Two major subsets of γδT lymphocytes are known, with peculiar localization and expression of a given variable δ chain: the Vδ1 subtype is mainly present in mucosal tissues, while the Vδ2 is the major subset circulates in the peripheral blood [188,189,190]. Different drugs can be utilized to expand γδT cells and improve their anticancer properties: in particular, pyrophosphate-containing compounds, such as isopentenyl pyrophosphate (IPP) and its isomer dimethylallyl pyrophosphate (DMAPP), have been proposed for cancer immunotherapy [191]. N-BPs, generally used to limit bone matrix catabolism, induce Vδ2T cell activation and proliferation, increasing the number of circulating γδT with anti-tumor activity [192,193,194]. Indeed, N-BPs, including alendronic, pamidromic, and zoledronic acid, are chemically stable analogues of inorganic pyrophosphates that inhibit the farnesyl pyrophosphate synthase (FPPS) of the mevalonate pathway and up-regulate IPP and DMAPP accumulation in healthy and neoplastic cells. The ability to recognize IPP by Vδ2T cells has been related to butyrophilin (BTN) 3A1 and 2A1 expression in target cells. These proteins are necessary for the binding and presentation of phosphates to Vδ2T cells, driving their activation [195,196,197]. For this immunostimulating property, besides the direct anti-cancer effects, different N-BPs have been employed in clinical trials [198,199,200].

### 5.2. Nanoformulations of N-BPs: Pharmacokinetics and Toxic Effects in Murine Models

The main pharmacokinetic features of N-BPs are their rapid clearance from the circulation and a specific bone tropism [200,201,202,203]. Indeed, these drugs are used to treat osteoporosis, due to their activity on osteocytes favouring bone matrix deposition [203,204]. This is the major drawback of N-BPs because the activation of Vδ2T cells could take place mostly within the bone, excluding other possible tumor sites when free drug formulations are used [200]. This property can be useful for the treatment of tumor bone metastasis, or primary bone tumors, such as MM [205]. On the other hand, encapsulation in liposomes (L) can increase the level of N-BPs at tumor sites other than bone (Figure 4) [42,206,207,208,209,210,211]. In addition,, intravenous administration of liposome-encapsulated zoledronate (L-ZOL) or alendronate (L-ALE) leads to consistent and high concentrations in the lung, liver, and spleen in murine models, with toxicity profiles different from their soluble forms. Indeed, L-ZOL was less tolerated than L-ALE and led to some sudden deaths in certain mouse strains, unlike free ZOL [209,210]. However, other reports stated that ZOL-containing self-assembly PEGylated NP or ZOL-encapsulated PEGylated liposomes do not exert any toxic effects in nude mice xenografts; these preparations did not affect mice weight or survival, nor did they induce necrotic effects or alterations in the inflammatory infiltrate in liver, kidney, and spleen (Table 3) [212].

These contrasting data on L-ZOL toxic effect can be explained considering the different composition of liposome preparations or the methods used for their generation. Indeed, L-ZOL that induced sudden mice deaths were composed of 1,2-distearoyl-*sn*-glycero-3-phosphocholine (DSPC):cholesterol:1,2-distearoyl-sn-glycero-3-phosphoethanolamine-N-amino(polyethylene (DSPE)-PEG2000 (55:40:2 mole ratio) and formulated using the Thin Film Hydration (TFH) method; on the other hand, nanoformulated ZOL NP were derived from a ZOL aqueous solution, calcium/phosphate nanoparticles (CaP NP), and cationic liposomes consisting of 1,2-dioleoyl-3-trimethylammonium-propane (DOTAP):cholesterol:DSPE-PEG (1:1:0.5) and were prepared by THF followed by extrusion [203,216,217]. The main toxic effect of liposome N-BPs preparations is on hematological parameters [209,210,211]. In particular, both L-ZOL and L-ALE induced neutrophilia and increased the percentage of neutrophils with a decrease of lymphocytes in the blood, but the effect of L-ZOL was stronger than that of L-ALE [209,210,211]. No evident tissue damages were found in heart, lung, liver, spleen, and kidney; thus, the molecular mechanism and the target of L-ZOL-induced mouse death is still to be identified. Whatever the reason for the contrasting safety profiles reported, the main obvious consequence is that each N-BPs nanoformulation should be assessed for its toxic effects. It remains to be determined whether murine models can be considered the most appropriate tool for this purpose, as they do not completely resemble the complexity of the human body [218].

### 5.3. Functional Behavior of N-BPs Nanoformulations: Direct and Indirect Anti-Tumor Effect

It has been reported that N-BPs nanoformulations can exert a direct effect on tumor cell growth [42,210,212,217,219,220]. A recent review well describes the pharmacological differences between some liposomal formulations and free N-BPs, as well as between non-targeted and targeted-N-BPs liposomes [210]. N-BPs encapsulated in different particles proved to affect the growth of breast and prostate carcinomas, gliomas and bone tumors [42,210,212,217,219,220]. Free N-BPs can lead to osteoclast apoptosis or pyroptosis, by acting on the FPPS and geranylgeranyl diphosphate synthase (GGPPS); N-BPs also trigger the osteoblast differentiation through the production of interferon (IFN) β in animal models [221,222]. Furthermore, in humans, N-BPs can display anti-tumor activity and concur with other treatments to a better clinical outcome [223,224]. Nanoformulated N-BPs can dramatically increase their localization and effect in tissues different from the bone; the main relevant question is whether and how the anti-tumor effect of free N-PBs diverges from differently nanoformulated N-BPs [209,210]. A marked alteration of the pharmacokinetic properties of N-BPs and a strong concentration of these drugs in liposomes are reported [209,210]. Liposomes composed of a FT-DSPE ligand can interact with FR expressed on tumor cells leading to a stronger uptake by tumor cells, compared to free N-BPs, which was accompanied by a 200 to 400 fold gain in anti-tumor effect, depending on the tumor cell line tested [225,226,227,228]. However, L-ZOL and L-ALE were not as efficient as their free counterpart, at least in vitro [210]. Self-assembled NP encapsulating ZOL can reduce the production of IL6, vascular endothelial growth factor (VEGF), fibroblast growth factor-2, and the CCL5 chemokine by MSC (Figure 4) [213]. These factors, produced during the crosstalk between MSC and the prostate cancercell line PC3, were involved in the clonogenic growth of PC3. Thus, NP loaded with ZOL can both inhibit the proliferation and migration of PC3 and influence stromal cells, suggesting a role for N-BPs in TME shaping. Accordingly, it has been recently reported that cytosine-phosphate-guanosine (CpG)-loaded metal-organic framework nanoparticles, loaded with ZOL for bone targeting, can induce M1 pro-inflammatory macrophage polarization both in vitro and in the intra-tibial murine model of breast cancer bone metastasis [229]. These findings are in agreement with the ability of free ZOL to reduce the production of transforming growth factor (TGF) β and increase IL15 secretion by MSC; this can favor the selection of Th1 pro-inflammatory T cells in NHL in vitro [230]. Thus, N-BPs can shape the TME towards an efficient immune response besides having a direct cytotoxic effect on tumor cells [189,231].

It has been shown that L-ALE and L-ZOL trigger Vδ2T cells to produce IFNγ and kill tumor target cells. However, liposome formulated N-BPs triggered Vδ2T cells to produce lower amounts of IFNγ and exert a less efficient cytotoxic activity than free N-BPs at the same molar concentration [209,210]. Nevertheless, the in vivo effect of L-ALE in combination with Vδ2T cells led to a significant reduction of the pseudo metastatic A375Pβ6 lung tumor in mice, while L-ALE or Vδ2T cells used alone showed marginal effects. No toxic effects were detected with several inoculations of L-ALE and Vδ2T cells, suggesting that the use of this combinatorial therapy is safe and effective [210]. FT-targeting of liposomal N-BPs was used to aggress ovarian tumors with expanded Vδ2T cells both in vitro and in vivo [232]. The cytotoxic activity elicited by pulsing tumor cells with L-ZOL or L-ALE was detected at a ten-fold lower amount of free N-BPs, indicating that triggering of Vδ2T cells can be more potent using an appropriate nanoformulation. The delivery of liposomal N-BPs can also benefit the EPR effect (see Section 4.2) [214]. EPR is triggered by the absence of lymphatic drainage at the tumor site, which further concentrates the liposome-formulated drug [231]. More recently, it has been shown that human radiolabeled Vδ2T cells can reach the xenogenic tumor in NOD SCID gamma (NSG) mice localizing at different sites, including the peritoneal cavity, subcutaneous tissue, and lung. This localization is not enhanced by the previous administration of L-ALE [233]; nevertheless, the combined immunotherapy with Vδ2T cells and L-N-BPs was effective in eliminating tumor cells and prolonging mice survival [209,210,224]. These findings suggest that N-BPs do not drive the localization of Vδ2T cells; thus, it is crucial to understand the molecular mechanisms that underlie the Vδ2T cell migration into the tumor to optimize their anti-tumor effect. In this context, we have recently reported that ZOL encapsulated in spherical polymeric nanoparticles (SPN) [215] display a strong efficiency in expanding Vδ2T cells from both the peripheral blood and neoplastic mucosa of patients suffering from CRC. These Exo-like NP were composed of Poly (D,L-lactide-co-glycolic) acid (PLGA), 1,2-dipalmitoyl-sn-glycero-3-phosphocholine (DPPC) and DSPE-PEG at 7.5:2.5 DPPC:DSPE-PEG molar ratio [214,215]. ZOL-charged SPN can stimulate Vδ2T cell expansion at a concentration about 300-fold lower compared to free ZOL. These ZOL nanoconstructs (Figure 4) sensitized both CRC cell line-derived spheroids and patient-derived CRC organoids to Vδ2T cell-mediated killing upon priming with a 20-fold lower amount compared to free ZOL [214]. These findings suggest that 3D cultured human primary CRC organoids could be a useful tool to study the functional behavior of Vδ2T lymphocytes and their efficiency in eliminating autologous tumor cells [214]. Notably, ZOL-charged SPN did not exert cytotoxic effects on peripheral monocytes or T lymphocytes at the molar concentrations used for triggering Vδ2T cells, supporting the idea that these nanoformulations can display a higher therapeutic index than free N-BPs drugs [214]. The main discoveries regarding direct and indirect effects of non-N- and N-BPs are summarized in Figure 5 [234,235,236,237,238,239,240,241,242,243,244,245,246,247,248,249,250,251,252].

## 6. Challenges for Clinical Implementation

According to experimental and preclinical studies, evidence confirms the positive application of EV/Exo as versatile drug carriers that could replace synthetic liposomes and nanocarriers, thanks to their natural role in tissue physiology, increased half-life, and low immune impact. On the other hand, due to their pleiotropic characteristics, NP are flexible tools for theranostic purposes, tuning their properties to gain advantages in the visualization of drug biodistribution and monitoring of drug release and efficacy. Nevertheless, clinical translation of EV and NP is still at the beginning, and further studies are needed to better shape and implement their performance. Strategies enhancing the ability of these agents to reach tumors by facilitating active targeting, combined with improved loading methods, are currently under investigation.

For EV, simple purification protocols and specific localization to tumor cells are still open issues. In addition, while lipophilic molecules can directly enter the EV lipid bilayer, hydrophilic molecules must be carried into EV by mechanical or chemical procedures that may reduce EV quality [200]. Bridging molecules, RNA aptamers, and chimeras are presently applied to optimize targeting and enhance the effectiveness of EV [69,76], specific engineering of EV surface remaining the most substantial approach.

As for NP, both size and shape seem to influence the ability to extravasate and reach the tumor site and the rate of drug delivery [105]. In addition, there is evidence that the EPR effect can be limited by abnormalities in tumor vasculature and extracellular matrix that impair effective delivery of NP-carried drugs [105,106]. A recent approach to face this problem is the particle replication in non-wetting templates (PRINT), whereby uniform NP can be assembled and customized for size and shape [105,106]. Modify drug delivery mechanism seems to be another interesting approach: for instance, physical stimuli, such as ultrasound, enzymes, light, magnetism, or hyperthermia, can trigger and enhance drug discharge from liposomes (reviewed in Li [105]). Redox, pH, and thermal controlled or enzyme-induced drug release can also be applied to most NP types [105]. One of the major advantages of dendrimers is represented by their branched conformation that allows easy loading of both hydrophobic and hydrophilic substances. In turn, dendrimers display a certain degree of toxicity towards normal cells; modifications of their surface groups with non-toxic reagents have been proposed to overcome the hematological toxicity of these NP [106].

In addition, for nanomedicine-based immunotherapy, the major challenge is the optimization of tumor targeting, drug delivery vs clearance and control of toxicity. In particular, to achieve an efficient and sustained anti-tumor immune response, controlled release of immunostimulating substances, together with anti-canacer drugs, combined with specific targeting are needed. This can be facedin part choosing local rather systemic administration of the therapeutic tool, including NP, when possible [105,106].

For all these reasons, despite the promising data of preclinical studies, clinical applications of EV/Exo and all the various types of NP are low in comparison with nanomedicine studies based on ADCs. Indeed, the clinical trial database (clinicaltrial.gov) reported 83 records for exosome-based diagnostic and therapeutic purposes in cancer and 204 active clinical trials for NP, among which only 11 regarding hematologic neoplasms. On the other hand, more than 250 active trials are present in the database searching for ADCs in solid cancers and more than 100 ongoing trials only for hematological malignancies. Antibody-driven targeting of NP and co-delivery of multiple components might represent possible solutions to bridge cancer therapy and induction of anti-tumor immune response at the site of the lesion. Thus, NP loaded with immunostimulant drugs and coated by specific therapeutic monoclonal antibodies or with antibodies directed to immuno-checkpoints might serve the purpose. We hope this scientific and technological effort will be applied in an increasing number of studies in the near future.

## Figures and Tables

**Figure 1 cancers-12-01886-f001:**
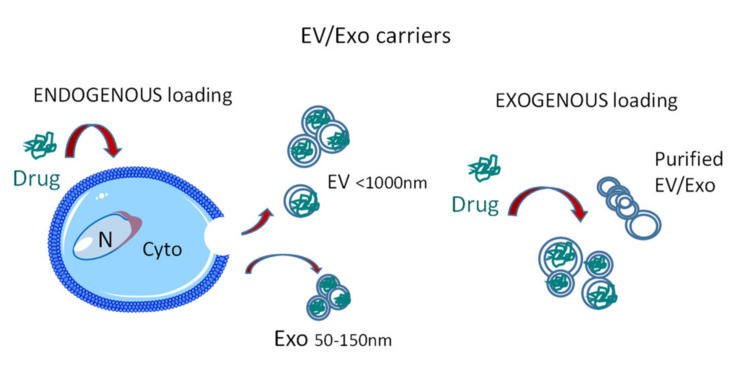
Extracellular vesicles (EV)/exosomes (Exo) as drug carriers: endogenous or exogenous loading. Drugs can be loaded into EV/Exo in two ways: (a) endogenous loading exploits EV/Exo-producing cells that are treated with the drug of interest, which is carried into EV/Exo inside the cell and eventually released as EV cargo in the extracellular milieu; (b) exogenous loading is obtained by manipulating isolated purified EV/Exo to carry the desired agent.

**Figure 2 cancers-12-01886-f002:**
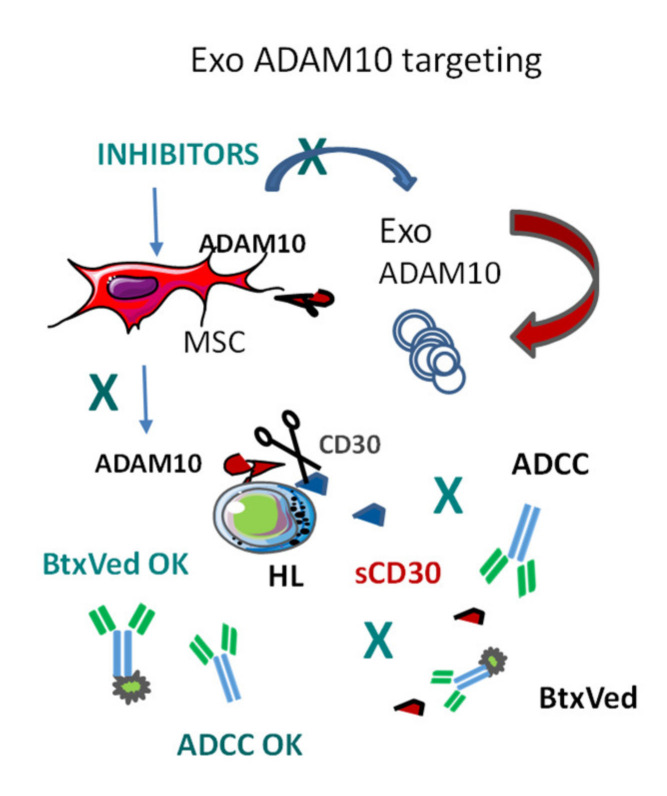
Regulation of immune response by Exo and endogenous loading of Exo to trigger anti-tumor immune response. In Hodgkin’s lymphoma (HL), A Disintegrin And Metalloprotease-10 (ADAM10) causes the shedding of CD30 that binds to the therapeutic anti-CD30 monoclonal antibodies (mAb) brentuximab-vedotin (BtxVed) impairing its cytotoxic effect on lymphoma cells and inhibiting antibody-dependent cellular cytotoxicity (ADCC) induced by iratumumab. ADAM10 can also be supplied by Exo from mesenchymal stem/stromal cells (MSC) to HL cells, amplifying CD30 shedding. When MSC are treated with ADAM10 inhibitors, the drugs are loaded in MSC-derived Exo, leading to the block of exosomal ADAM10 enzymatic activity and restoring both ADCC and BtxVed therapeutic effect.

**Figure 3 cancers-12-01886-f003:**
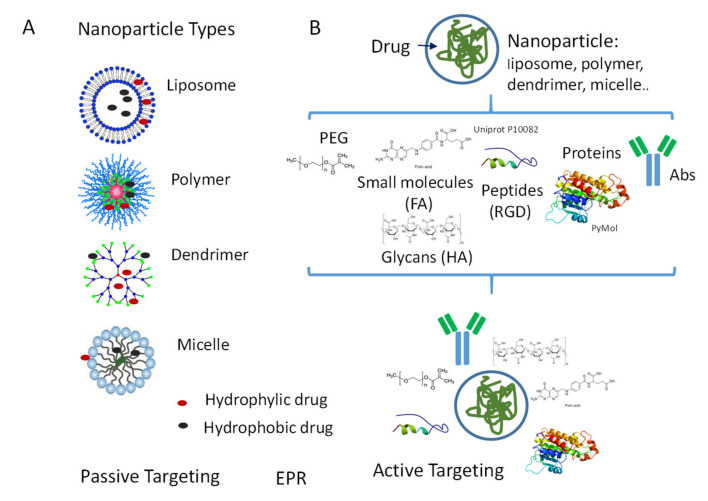
Nanoparticles (NP) types: passive and active targeting. (**A**) Schematic representation of different NP types: liposomes, polymeric NP, dendrimers, and micelles. Passive targeting: accumulation of a drug-driven by NP to the site of the lesion. EPR: “enhanced permeation and retention effect”, which allows NP to pass tumor vascular endothelium. (**B**) Active targeting: surface modifications of NP responsible for the increase of circulation half-life and selectivity of delivery. Examples of such modifications: polyethylene glycol (PEG)ylation, coating with sugars, small molecules, and peptides. FT: folate; HA: hyaluronic acid. Peptides containing sequences involved in internalization (such as RGD) improve drug uptake. Conjugation with specific therapeutic antibodies (Abs) will confer to NP high specificity of targeting.

**Figure 4 cancers-12-01886-f004:**
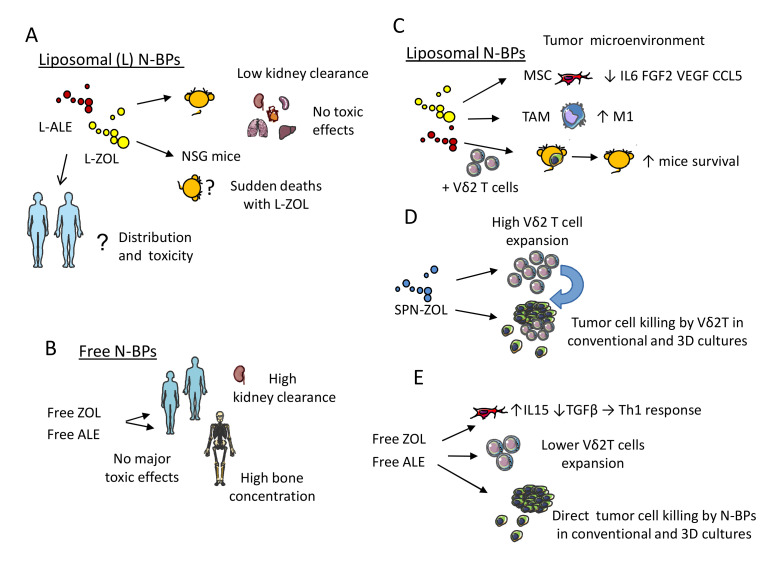
Pharmacokinetics, direct and indirect effects of N-BPs nanoformulations.(**A**) Liposomal (L) formulations of zoledronate (L-ZOL) or alendronate (L-ALE) in murine models: localization, clearance, and effects, including sudden deaths of non-obese diabetic severe combined immunodeficient (NOD SCID) gamma (NSG) mice following L-ZOL administration. No data are reported for L-ZOL and L-ALE in humans. (**B**) Free ZOL and free ALE distribution, effects, and clearance in humans. (**C**) Liposomal N-BPs interaction with tumor microenvironment. MSC: mesenchymal stromal cells; TAM: tumor-associated macrophages; M1: type 1 macrophages; IL6: interleukin-6; FGF2: fibroblastic growth factor-2; VEGF: vascular endothelial growth factor; CCL5: chemokine (C–C) ligand 5.L -ALE followed by Vδ2 T cells increases the survival of immunodeficient mice bearing pseudo-metastatic lung tumor. (**D**) SPN-ZOL: spheric polymeric nanoparticles charged with ZOL. (**E**) Free ZOL and free ALE reduce the immunosuppressive cytokine TGFβ (transforming growth factor-β) and increasing the immunostimulant growth factor IL15 (interleukin 5), leading to the elimination of tumor cells in conventional and 3D culture systems.

**Figure 5 cancers-12-01886-f005:**
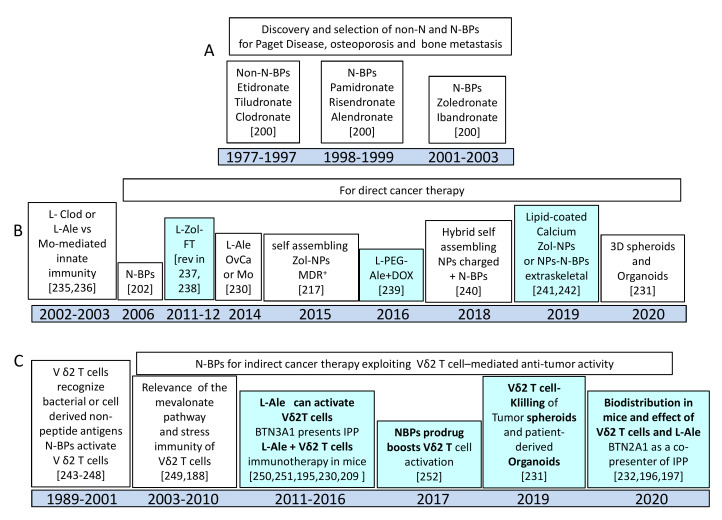
Timelines of the most important findings on non-amino-BPs and N-BPs: direct and indirect anti-tumor effects of N-BPs. In light blue and bold, the more relevant discoveries are highlighted. Zoledronate is the most potent amino bisphosphonate available. (**A**) Main features of non-amino and amino bisphosphonates are reviewed in Ref. [200]. (**B**) Non-N and N-BPs employed as free drugs or as liposome (L) or nanoparticles (NPs) to inhibit tumor cell growth. (**C**) Vδ2 T cell responses to N-BPs as either free drugs or L or NPs formulations. Some of the main discoveries regarding the presentation of isopentenyl pyrophosphates (IPP) are indicated to point out the molecular mechanisms underlining the N-BPs-mediated effects. For each major finding, appropriate references are inserted. Abbreviations: Clod: Clodronate; Ale: Alendronate; Zol: Zoledronate; FT: Folate; PEG: Polyethylene Glycol; MDR: multidrug resistance; DOX: Doxorubicin; Mo: monocyte; BTN: Butyrophilin subfamily.

**Table 1 cancers-12-01886-t001:** Characteristics, source, and therapeutic use of EV and Exo.

Vesicle Type ± Cargo	Source	Size	Experimental Model/ Condition/Disease	Molecular/Cellular/ Tissue Target	Therapeutic Potential	Ref.
EV/Exo native/ autologous/ allogenic	Tissue/stem cells/MSC	30 nm–1 µm 8–12 nm	Tissue injury/tissue repair cardioprotection/ neurodegeneration/ immune response	Injured tissues/stem cells/inflammatory cells/immune cells	Cardiac-lung-liver repair/regeneration/ immune regulation	[17,18,19,20,21,25,26,27,28,35]
EV native	Adipose tissue	50 nm–1 µm	Type 2 diabetes	Vasculature/ immune cells	Metabolic/ immune regulation	[47]
Exo+curcumin	EL4 mouse lymphoma/RAW 264.7 murine macrophage cell line and others	30–100 nm	In vivo LPS-induced septic shock mouse model	Activated myeloid cells/inflammatory cells	Improve anti-inflammatory activity of curcumin	[51]
Exo+paclitaxel	PC3/LNCaP prostate tumor cell lines	50 nm–1µm	In vitro model of EV uptake in prostate tumors	PC3/LNCaP prostate tumor cell lines death	Improve paclitaxel cytotoxicity	[50]
AA-PEG-exo PTX	Macrophages	30–100 nm	In vivo mouse model pulmonary metastases	Metastatic cell death	Improve therapeutic outcome	[67]
EV+cc-siRNAs	Neuro 2A cell line/mouse DCs	50 nm–1 µm	Transfection of cc-siRNA loaded-EV in HEK293/ Neuro2A murine neuroblastoma/SH-SY5Y human neuroblastoma/ GM04281 Huntington’s disease fibroblasts	Human antigen R Tumor cell death	Specific delivery of anticancer drugs to neuroblastoma and glioma	[52]
Exo-Dox	MSC transduced with HER ligands	30–100 nm	In vitro uptake by HER^+^ BT474/SKBR3 breast tumor cell line	Specific targeting of doxorubicin to HER^+^ breast tumor cells	Improve doxorubicin efficacy	[53]
A33Ab-US/Exo/Dox	LIM1215 cells	30–100 nm	CRC cell lines/in vivo CRC mouse model	Tumor growth inhibition	Improve doxorubicin efficacy/better outcome/survival	[70]

Abbreviations: MSC: mesenchymal stem/stromal cells; AA-PEG-exoPTX: Exo vectorized with AA-PEG loaded with paclitaxel; EV+cc-siRNAs: EV+ cholesterol-conjugated small interfering RNAs; DCs: dendritic cells; HEK293: human embryonic kidney cell line; Exo-Dox: Exo encapsulated doxorubicin; A33 Ab-US/Exo/Dox: Exo A33 Ab-functionalized loaded with doxorubicin; CRC: colorectal carcinoma.

**Table 2 cancers-12-01886-t002:** NP types: characteristics, biodistribution, benefits, and drawbacks.

NP Type	Size	Biodistribution	Benefits	Drawbacks	Ref.
Liposomes	200 nm–1 µm	Variable, poorly predictable	Good biocompatibility and stability, reduced drug toxicity, reduced drug degradation	Variable and slow drug delivery, liposome degradation	[105,107]
Natural polymers (peptides, glycans)	20–300 nm	Good distribution and organ targeting	Good biocompatibility, optimal active drug transport	Inflammatory response, degradation	[105,109,110]
Synthetic polymers (PLA, PLGA)	50–200 nm	EPR effect, organ-specific targeting	Improved drug solubility, reduced toxicity, prolonged circulation time, thermosensitive delivery	Inflammatory response, immune reaction, degradation, cytotoxicity	[106,111,112,113,114]
Dendrimers	50–100 nm	Optimal distribution and kidney elimination	Branches carrying the drug, controlled release, self-assembly	Immunological reaction, hematological toxicity, cytotoxicity	[105,109,120]
Micelles	20–150 nm	Optimal distribution	Good biocompatibility and passive targeting, good drug solubilization and reduced degradation, self-assembly	Possible cytotoxicity	[111,112,121]
Inorganic NP (iron oxides, gold NP, carbon nanotubes)	40–100 nm	Good distribution and tissue localization	Theranostic use, enhanced tumor imaging and radiotherapy	Metal toxicity, storage	[123,124,125,126,127]

Abbreviations: NP: nanoparticles; PLA: polylactic acid; PLGA: lactic-co-glycolic acid; EPR: enhanced permeability and retention.

**Table 3 cancers-12-01886-t003:** Features of free nanoformulated aminobisphosphonates (N-BPs) and nanoparticles (NP) charged with N-BPs.

N-BPs Form	Size nm	Cellular Biodistribution	In Vivo Biodistribution	In Vitro Functional Effects	In Vivo Functional Effects	Ref.
Free alendronic acid or free zoledronic acid	NA	Passive penetration and inhibition of FPPS, Generation of IPP and DMAPP	High plasma and renal excretion (min) and high bone concentration and release along time half-life (years)	Anti-tumor toxic effects triggering Vδ2 T cells	Potent anti-osteoporotic Effects Use for bone metastasis treatment	reviewed in [200]
high phase-transition phospholipids, cholesterol and pegylated liposomes for alendronic acid	85–100 nm Neg charged	High cellular penetration	High Plasma half-life (18 h) Low renal excretion High extraskeletal tissue	Anti-tumor toxic effects triggering Vδ2 T cells	Murine models Anti-tumor effects Synergistic effects with Vδ2 T cell administration	[200,209,210,211]
high phase-transition phospholipids, cholesterol and pegylated liposomes for zoledronic acid	85–100 nm Neg charged	High cellular penetration	High Plasma half-life (18 h) Low renal excretion High extraskeletal tissue	Anti-tumor toxic effects triggering Vδ2 T cells	Murine models anti-tumor effects and sudden death	[200,209,210,211]
ZOL-containing self-assembly PEGylated nanoparticles (NPs) or ZOL-encapsulating PEGylated liposomes DOTAP/chol/DSPE-PEG	147–245 nm +1.8–17.5 mV ζ pot	Not performed specific studies	Not performed specific studies	Not performed specific studies	Potent anticancer effect in murine model of prostate adenocarcinoma	[212]
PEGylated ZOL- charged NPs with CaPZ NPs and DOTAP/chol/DSPEG2000	147 nm +18 mv ζ pot	Strong effects on mesenchymal stromal cells and tumor cells	Not performed specific studies	Not performed specific studies	Not performed specific studies	[213]
Spheric polymeric nanoparticles DPPC:DSPE-PEG7.5:2.5 molar ratiocharged with zol	171 nm –43 mV ζpot	Penetration and generation of IPP in tumor cells and Mo, no toxicity on Mo	Not assessed as Zol charged NPs	Potent boost of Vδ2 T cell proliferation and cytolysis of 3D CRC spheroids and patient-derived organoids	Not assessed as Zol charged NPs Potent antitumor effects on orthotopic glioblastoma multiforme in murine models as docetaxel- and diclofenac-charged NPs	[214,215]

Abbreviations: FPPS: farnesyl pyrophosphate synthase; IPP: isopentenyl pyrophosphate; DMAPP: Dimethyl-allil pyrophosphate; DPPC:DSPE-PEG: 1,2-dipalmitoyl-sn-glycero-3-phosphocholine (DPPC), 1,2-distearoyl-sn-glycero-3-phosphoethanolamine-N-[succinyl(polyethylene glycol)-2000] (DSPE-PEG),DOTAP: 1,2-dioleoyl-3-trimethylammonium-propane chloride. Mo: monocytes; CRC: colorectal carcinoma.

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
