# Peer review of "Cancer Nanomedicine Special Issue Review Anticancer Drug Delivery with Nanoparticles: Extracellular Vesicles or Synthetic Nanobeads as Therapeutic Tools for Conventional Treatment or Immunotherapy"

_cancers, 2020, doi:10.3390/cancers12071886_

Round 1

Reviewer 1 Report

The authors have presented a review article discussing delivery of anti-cancer drugs using extracellular vesicles and nanoparticles. The information presented describes sources of EVs and multiple types of nanoparticles, different methods for loading as well as modifications that can be performed to increase therapeutic efficacy. Multiple examples have been shown as supporting evidences with special attention to anti-tumor immune response. Overall, the manuscript is interesting and compiles the latest information on drug delivery using nanoparticles which can be useful to the readers. The following suggestions can be implemented -

  1. Include a paragraph about EVs and exosomes in the Introduction.
  2. Kindly include challenges of cargo loading in EVs and NP, pros and cons of multiple methods used for drug loading, efficiency etc. Sustained delivery of drugs is another method known in the field. Please add information for this topic.
  3. Section 2.2 and 3.1 can be combined since there is repetitive information.
  4. Include a table for types of NPs to include- size differences, efficiency, half -life, biodistribution, toxicity etc.
  5. Include References from line 381-391 on page 10.
  6. Some information is redundant in the manuscript. For example- PEGylation of EVs and NPs, Hyaluronic acid -CD44 targeting, Paclitaxel loaded NPs. I suggest the authors to remove information that is repetitive.

Author Response

Reviewer 1:

  1. Include a paragraph about EVs and exosomes in the Introduction.

This paragraph has been added to the Introduction (page 3, lines 73-81)

  1. Kindly include challenges of cargo loading in EVs and NP, pros and cons of multiple methods used for drug loading, efficiency etc. Sustained delivery of drugs is another method known in the field. Please add information for this topic.

This has been added to the last new paragraph “Challenges for clinical implementation”, to harmonize the topic with the answer to other issues raised by Reviewer 2. In particular, see pages 21,22 lines 667-686.

  1. Section 2.2 and 3.1 can be combined since there is repetitive information.

This chapter has been rewritten and section 3.1 combined to 2.2 (pages 3-7, lines 90-254)

  1. Include a table for types of NPs to include- size…, efficiency, half-life, biodistribution, toxicity etc.

This table (Table II)  has been prepared as also requested by Reviewer 2.

  1. Include References from line 381-391 on page 10. This paragraph is free of references because it is a figure legend.
  2. Some information is redundant in the manuscript. For example- PEGylation of EVs and NPs, Hyaluronic acid -CD44 targeting, Paclitaxel loaded NPs. … remove information that is repetitive.

We have removed some redundant information. Nevertheless, CD44 and PD1-PDL1, as well as several hints at PEGylation are still present referred to either EV or NP or ZOL, since this type of targeting is used in either case with some peculiarities and illustrate distinct phenomena or different trials.

Reviewer 2 Report

The review entitled “Anticancer drug delivery with nanoparticles: extracellular vesicles or synthetic nanobeads as therapeutic tools for conventional treatment or immunotherapy.” by Poggi et al. discuss the state-of-the-art in nanotechnology applied for theranostic use. The review works on a very broad topic, but the authors mainly focus it on EV as the most promising nanocarrier for clinical translation. In a first stage, the authors explain, from an academic point of view, EV sources and their modification for clinical purposes. Then, there is a discussion of the potential of Exo as drug carriers to enhance anti-tumor immune response. In a second part, the authors report the very-well known expectations that the use of nanotechnology raises in the field of oncology. More deeply, they explain the virtues that synthetic nanocarriers have brought to the field regarding pharmacokinetic, reduction of adverse events and, even, their potential use in the treatment of different malignancies. Briefly, the authors break down the most representative and positive findings in the field with not critical comments. Of note, in both sections the authors set outs the progress that has been achieved in clinical trials. Finally, the authors configurate a section dedicates to aminobisphosphonates as inmunostimulators.

The work is very well written and very well documented, the revision is almost encyclopaedic, in line with other excellent works reported for the corresponding author. They have accomplished an exhaustive revision in the field, from which I can only congratulate them for the hard work done. However, the work is not enjoyable to read. There is so many information in each section regarding findings in the matter, but there is not selection of which are the most important to discuss. I am afraid, the review needs a bit of work to be published in Cancers. Please, follow my suggestions:

-The section 2.2 of the review should be constrained. This review is not dedicated to methodologies.

-The authors should prepare a Table from which readerships could easily find the information and compare results and findings in the matter under discussion. I propose three tables, the first one only dedicate to EV, other one to Nanoparticles in general, and finally the latter regarding aminobisphosponates. In this context, I would appreciate whether the authors report an extra Figure representing a timeline with the most important findings regarding aminobisphosphonates.  

-It will be appropriate to read not only virtues about the use of nanotechnology. Nanomedicine raises high expectations for millions of patients as it can provide better, more efficient, and affordable healthcare, and has the potential to propose novel therapeutics for the treatment of solid tumors. However, the number of nanomedicines in clinical trials are not significant in comparison with ADCs.

-There is a vast amount of information in  figure captions. This is quite extraordinary for a review . I suggest incorporating it along the discussion in some way.

- With strategies enhancing the ability of these agents to reach tumors by facilitating active targeting, combined with improved uniform manufacturability, it is anticipated that there will be an increase in the interest of this family of agents for clinical development. A section dedicated to “Challenges for clinical implementation” at the end of the manuscript should be welcome.

Author Response

Reviewer 2:

-The section 2.2 of the review should be constrained….Section 2.2 has been shortened and merged with section 3.1 as also requested by reviewer 1 ((pages 3-7, lines 90-254)

-………I propose three tables, the first one only dedicate to EV, other one to Nanoparticles in general, and finally the latter regarding aminobisphosponates. In this context, I would appreciate whether the authors report an extra Figure representing a timeline with the most important findings regarding aminobisphosphonates.

Table I on EV/Exo and Table II on NP characteristics have been prepared. Data and information on free and naformulated N-BPs have been included in Table III and Figure 5. This figure depicts the timelines of the main findings on non-N bisphosphonates and N-BPs as nanoformulations and N-BPs effects on Vδ2T cells, with particular regard to immune stimulation and NP applications.

-It will be appropriate to read not only virtues about the use of nanotechnology. ………. However, the number of nanomedicines in clinical trials are not significant in comparison with ADCs…….

We agree with the reviewer and this issue has been addressed and included in the new suggested paragraph 6, “Challenges and clinical implementation”, pages 22,23 lines 667-713.

-There is a vast amount of information in figure captions. This is quite extraordinary for a review. I suggest incorporating it along with the discussion in some way.

Legends to figures have been shortened as suggested.

- …….A section dedicated to “Challenges for clinical implementation” at the end of the manuscript should be welcome.

A new paragraph (6) entitled “Challenges for clinical implementation” and dedicated to the topic suggested has been included (pages 22,23, lines 667-713)

Round 2

Reviewer 2 Report

Thank you very much for the effort. The authors have followed my recommendations. I suggest to publish the work in present form.